# Mitochondrial Homeostasis in VSMCs as a Central Hub in Vascular Remodeling

**DOI:** 10.3390/ijms24043483

**Published:** 2023-02-09

**Authors:** Yi Xia, Xu Zhang, Peng An, Junjie Luo, Yongting Luo

**Affiliations:** Department of Nutrition and Health, China Agricultural University, Beijing 100193, China

**Keywords:** vascular remodeling, mitochondria, VSMC, fission, fusion, mitophagy, mtDNA

## Abstract

Vascular remodeling is a common pathological hallmark of many cardiovascular diseases. Vascular smooth muscle cells (VSMCs) are the predominant cell type lining the tunica media and play a crucial role in maintaining aortic morphology, integrity, contraction and elasticity. Their abnormal proliferation, migration, apoptosis and other activities are tightly associated with a spectrum of structural and functional alterations in blood vessels. Emerging evidence suggests that mitochondria, the energy center of VSMCs, participate in vascular remodeling through multiple mechanisms. For example, peroxisome proliferator-activated receptor-γ coactivator-1α (PGC-1α)-mediated mitochondrial biogenesis prevents VSMCs from proliferation and senescence. The imbalance between mitochondrial fusion and fission controls the abnormal proliferation, migration and phenotypic transformation of VSMCs. Guanosine triphosphate-hydrolyzing enzymes, including mitofusin 1 (MFN1), mitofusin 2 (MFN2), optic atrophy protein 1 (OPA1) and dynamin-related protein 1 (DRP1), are crucial for mitochondrial fusion and fission. In addition, abnormal mitophagy accelerates the senescence and apoptosis of VSMCs. PINK/Parkin and NIX/BINP3 pathways alleviate vascular remodeling by awakening mitophagy in VSMCs. Mitochondrial DNA (mtDNA) damage destroys the respiratory chain of VSMCs, resulting in excessive ROS production and decreased ATP levels, which are related to the proliferation, migration and apoptosis of VSMCs. Thus, maintaining mitochondrial homeostasis in VSMCs is a possible way to relieve pathologic vascular remodeling. This review aims to provide an overview of the role of mitochondria homeostasis in VSMCs during vascular remodeling and potential mitochondria-targeted therapies.

## 1. Introduction

Vascular remodeling refers to the structural and functional changes in the vasculature in response to injury, ageing and diseases [1]. This process is initially adaptive but is likely to develop to be pathogenic and unrecoverable under sustained disturbance. The vascular wall is typically formed by endothelial cells, smooth muscle cells and fibroblasts. During pathogenic remodeling, vascular cell proliferation, death, migration and phenotypic transformation are involved in the response to oxidative stress, inflammation, vascular calcification and other stimuli [2]. Pathogenic vascular remodeling contributes to numerous cardiovascular diseases. Here, we will focus on vascular remodeling in the aorta and its associated diseases, especially hypertension, atherosclerosis and arterial aneurysm [3,4].

Hypertension is an inward eutrophic remodeling disease involving the artery and is pathophysiologically characterized by increased vascular resistance, which is determined by a reduced vascular diameter due to increased vascular contraction and arterial remodeling [5]. Vascular smooth muscle cells (VSMCs) are one of the mediators of complex systems, including the renin–angiotensin–aldosterone system, the sympathetic nervous system, immune activation and oxidative stress, which regulate hypertension. The increase in intracellular calcium ion concentration leads to VSMC contraction, which promotes vascular contraction, triggering acute and rapid adaptation of blood vessel diameter [5]. This adaptive diameter change becomes maladaptive under chronic pathological conditions, contributing to vascular remodeling and stiffness [6]. Physiologically, VSMCs are of contractility phenotype and exhibit low levels of proliferation. In contrast, VSMCs can dedifferentiate or undergo a phenotypic transformation in the pathological status [5]. In hypertension, VSMC phenotypic switch contributes to vascular dysfunction and arterial remodeling. Adaptive vascular remodeling is responsible for vascular repair in the early stage of hypertension when the proliferation of VSMC is strictly regulated. However, with the aggravation of hypertension, VSMC proliferation is out of control and the dedifferentiated VSMC accumulates on the vascular wall, leading to media thickening, intima hyperplasia and vascular sclerosis [7], which indicates that pathological vascular remodeling develops. During the process, diverse signal mediators, such as AngII, endothelin 1 (ET-1) [8] and aldosterone [9], initiate vascular remodeling of VSMCs.

Atherosclerosis is a chronic disease characterized by lipid-laden plaques and damaged arterial walls [10]. Unlike hypertension, atherosclerosis is associated with a positive vascular remodeling in the early stage and the lumen is dilated to stabilize the blood flow. As the disease progresses, negative remodeling leads to lumen stenosis and plaque blockage with compensatory imbalance [11]. Furthermore, pathological vascular remodeling can cause postoperative lumen stenosis, graft lumen lesions and post-lesion restenosis [12]. VSMCs, located in the medial layer of the vascular wall, participate in atherosclerosis and invade the fibrous cap and plaque center [10]. With the stimulation of inflammation or lipids, the damaged vascular wall exposes the media layer together with VSMC migration to the intima layer. These VSMCs accumulate and abnormally proliferate in the intima [13]. Otherwise, the pathological environment stimulates VSMCs’ phenotypic switch from contraction to synthetic, during which some cells are transformed into dedifferentiated states [13], associating with lumen thickening and plaque stabilization.

Arterial aneurysm and arterial dissection are life-threatening diseases characterized by structural vascular wall changes [14]. The integrity of the aortic wall can be damaged by factors such as the degradation of the extracellular matrix and inflammation. Together with the impact of blood flow, the vascular wall is prone to tear, leading to blood vessel rupture or the formation of a false lumen between the intima and the media layer [15]. There are some typical chronic aortic dissection pathological features: (1) increased aortic diameter, the aorta dilated quickly at first, then slowly; (2) thrombus slowly forms in the false lumen; (3) the dissection flap thickened initially, then deteriorated, straightened and lost its activity; and (4) there was significant vascular remodeling at the beginning, and as the disease aggravates, the remodeling slowed down and translated into fibrosis [16]. During the process, elastin fragments and fibrosis rating in medium layer are crucial markers of dissection progression. Damage to VSMCs leads to increased collagen deposition and fibrosis, whereas loss of VSMCs’ nuclei leads to necrosis of the medium layer [16].

As a critical component of the intima-medial of the vessel wall, VSMCs are mainly responsible for the contraction and relaxation of the vessel to control blood pressure and blood flow [17]. In vascular remodeling, VSMCs are characterized by hypertrophy, proliferation, phenotype transformation, migration, senescence and excess extracellular matrix [18,19]. In atherosclerosis, VSMCs migrate from the media into the intima and proliferate with the induction of platelet-derived growth factors [11]. Intimal hyperplasia is a severe consequence of VSMC proliferation, leading to luminal stenosis and endangering patients undergoing revascularization [20]. The abnormal status of VSMCs is tightly related to vascular growth factors, vasoactive substances, hemodynamics, oxidative stress and inflammatory factors [21,22].

VSMCs undergo contraction to regulate blood vessel width and blood pressure. Most of the energy for muscle contraction is provided by mitochondria [23]. Mitochondria not only produce ATP but also provide various metabolic substrates through the respiratory chain. Thus, the quality of mitochondria is crucial to keep VSMCs healthy. The morphology and function of mitochondria in VSMCs were changed responding to vascular injury [24]. Mitochondrial homeostasis is maintained by mitochondrial biogenesis, fission and fusion, and mitophagy. The disruption of these processes and the mitochondrial genome may affect mitochondrial quality and functions and are considered to promote VSMC-mediated vascular remodeling [25,26]. For instance, in both human and rodent atherosclerosis, mitochondrial respiration in VSMCs is impaired, resulting in increased glycolytic activity and decreased OXPHOS. Restoring mitochondrial function in VSMCs can limit or even reverse cardiovascular diseases [27,28].

In this article, we will focus on how mitochondrial homeostasis in VSMC drives the development of vascular remodeling and the corresponding treatments for restoring mitochondrial function.

## 2. Mitochondrial Biogenesis and Vascular Remodeling

Mitochondrial biogenesis indicates the generation of new mitochondria and is regulated to adapt to cell energy demands. The process is highly regulated by proliferator-activated receptor-γ coactivator 1α (PGC-1α). Diverse pathways are involved in regulating the expression of PGC-1α including receptor tyrosine kinases (RTKs), cyclic guanosine monophosphate (cGMP), AMPK and sirtuin 1 (SIRT1). PGC-1α can regulate the expression of nDNA-encoded OXPHOS components and mitochondrial DNA (mtDNA) by mitochondrial transcription factor A (TFAM) (Figure 1) [29].

PGC-1α plays a key role in SMC oxidative stress, apoptosis, inflammation and cell proliferation, which are relevant to the progression of vascular disease. Excessive proliferation of SMC contributes to the incidence of atherosclerosis, restenosis and pulmonary hypertension [30]. Serine 570 phosphorylation of PGC-1α stimulated by angiotensin II (Ang II) leads to decreased catalase expression and thus increases ROS levels in VSMCs and subsequent vascular hypertrophy [31].

Mitochondrial dysfunction of VSMCs can drive the aortic aneurysm in Marfan syndrome (MFS). In the aortas of MFS patients, all mitochondrial complex subunit expressions are reduced as are the genes related to mitochondrial function and mitochondrial biogenesis [32]. In non-hereditable aortic aneurysms and dissections induced by atherosclerosis, there is decreased mitochondrial respiration and a decrease in mitochondrial proteins in vascular smooth muscle cells from murine and human aortic aneurysms [33]. Mitochondrial biogenesis is also impaired in synthetic SMCs of abdominal aortic aneurysm patients [34].

## 3. Mitochondrial Fusion and Fission of VSMC in Vascular Remodeling

Mitochondrial dynamics encompass the processes of fusion, fission and selective degradation [24]. Studies have shown that mitochondrial dynamics play a key role in VSMC migration. Suppressing mitochondrial fission induced by platelet-derived growth factor (PDGF) limits VSMC migration and pathological intimal hyperplasia by altering mitochondrial energetics and ROS levels [35]. In addition to energy supply, mitochondrial morphology also influences cell migration [35]. Mitochondrial fission and fusion control the size, number and shape of mitochondria [24]. Shortened mitochondria promote cell migration without stimulation of PDGF [35], whereas elongated mitochondria correlate with more efficient ATP production [24].

Mitochondrial dynamics also affect the proliferation of VSMCs [36]. In a model of pulmonary hypertension, excessive mitochondrial fission induces the proliferation of pulmonary artery smooth muscle cells (PASMCs) [37]. Mdivi-1, a selective dynamin-related protein 1 (DRP1) inhibitor, can block mitochondrial fission leading to cell cycle arrest in the G2/M phase and reduced PASMC proliferation [27]. Several studies have indicated that chronic hyperglycemia promotes reactive oxygen species (ROS) production and VSMC proliferation, which aggravates vascular remodeling [38,39]. Oxidative stress caused by hyperglycemia stimulates mitochondrial fragmentation by increasing mitochondrial fission and decreasing fusion; activation of DRP1 is involved in this process [40]. Abnormal migration and proliferation of VSMCs are a feature of intimal hyperplasia [41]. Intimal hyperplasia is a chronic structural lesion that develops after vessel wall injury that leads to luminal stenosis and occlusion [41].

VSMC senescence leads to vascular ageing, which is associated with various vascular or metabolism diseases, such as hypertension, atherosclerosis and diabetes [42]. Dysregulation of mitochondrial dynamics is associated with the senescence of smooth muscle cells by producing excessive ROS [43,44]. Abdominal aortic aneurysm (AAA) is induced by mitochondrial reactive oxygen species (mtROS), which can be elevated by mitochondrial fission [45]. Transcriptional factor Kruppel like factor 5 (KLF5) can rescue AAA through the following pathways: (1) binding to the promoter of eukaryotic translation initiation factor 5a (eIF5a) and activating eIF5a transcription, which protects mitochondrial integrity by interacting with mitofusin 1 (MFN1) [44]; and (2) inhibiting mitochondrial fission to reduce mitochondria-derived ROS and prevent VSMC senescence [8]. For mitochondria, chronic stimulation, including angiotensin II (Ang II) and oxidative stress and inflammation, can disrupt the mitochondrial dynamic balance and promote mitochondrial fission and fragmentation, which lead to the accumulation of mtROS and VSMC ageing [46].

The phenotypic transition of VSMC from a contractile to synthetic phenotype occurs in vascular remodeling, especially in atherosclerosis and coronary heart disease [47,48,49]. The mitochondrial structure can regulate the phenotype and metabolic characteristics of VSMCs [28]. It has been reported that PDGF is involved in the phenotypic transition of VSMC in a mitochondrial-dynamics-dependent manner [28]. In vascular calcification, AMPK activation promotes mitochondrial fusion by down-regulating DRP1 [50]. During this process, VSMC osteochondrogenic transformation is inhibited and vascular calcification is alleviated. In the pathological state, the phenotypic switch of VSMCs correlated with the fusion and fission of mitochondria [28].

GTP-hydrolyzing enzymes of the dynamin superfamily are the key mediators of mitochondrial fusion and fission. MFN1, mitofusin 2 (MFN2) and optic atrophy protein 1 (OPA1) are crucial for mitochondrial fusion, whereas the center of mitochondrial fission is DRP1, which uses GTP hydrolysis to perform mechanical work on lipid bilayers [50]. Angiotensin II (AngII), inflammation, oxidative stress and other stimuli induce vascular-remodeling-related diseases with increased fission and reduced fusion and result in more mitochondria in the form of short tubules or spheres [14,45]. In addition to affecting mitochondrial morphology, mitochondrial fusion/fission homeostasis is closely related to mitochondrial function, including maintaining the cellular redox state and energy supply [51]. Mitochondrial fragmentation and mitochondrial dysfunction lead to mtROS accumulation and ATP level decline. The pharmacological inhibitor of DRP1 (mdivi-1) can inhibit mitochondrial fission [16] and activate the AMPK pathway to play a protective role in mitochondrial integrity [50]. Elf5a restrains the fragment of mitochondria by interacting with MFN1 [44]. Thus, the mitochondrial dynamics of VSMCs plays a vital role in vascular remodeling by affecting the energy metabolism and phenotypic transition of VSMCs (Figure 2).

## 4. Mitophagy of VSMC in Vascular Remodeling

Mitophagy is an autophagic phenomenon targeting damaged or redundant mitochondria to ensure proper mitochondrial quality control [25,52]. Hypoxia, inflammatory stimulation, insulin resistance and oxidative stress lead to mitochondrial damage in VSMCs, and mitophagy can eliminate damaged mitochondria to protect VSMC (Figure 3). PINK1/Parkin-mediated mitophagy obliterates the dysfunctional mitochondria derived from mitochondrial fission [53]. Although mitophagy typically plays a positive role in VSMC homeostasis, its effect on VSMC proliferation remains controversial. VSMC proliferation is a critical process in arteriosclerosis. Several studies have shown that mitophagy promotes VSMC proliferation [54]. Atherosclerosis is characterized by the accumulation of lipids, which evokes Apelin-13 expression [54]. Apelin-13 activates a PINK1/Parkin signal that initiates mitophagy in VSMC. Meanwhile, mitochondria dynamics assists mitophagy to accelerate VSMC proliferation through up-regulated DRP1 and down-regulated MFN1/2 and OPA2 [54]. VSMC proliferation is mainly determined by mitochondrial fission and fusion, wheras mitophagy only plays an auxiliary role. Mitophagy can repair mitochondrial function to inhibit VSMC proliferation. Once the level of damaged mitochondria exceeds the repair capacity of mitophagy, the proliferation of VSMC will increase simultaneously, which may explain the contradictory role of mitophagy in VSMC proliferation.

Mitophagy reduces the release of cytochrome C from dysfunctional mitochondria and other apoptotic factors to activate cell death pathways [55]. Defective mitophagy leads to the accumulation of fragmented mitochondria with reduced bioenergetic efficiency and oxidative stress [56]. Mitophagy protects VSMC against oxidative stress and resists cell apoptosis [56]. Atherosclerosis and vascular calcification are often accompanied by VSMC apoptosis. The metabolism of VSMC is altered in these diseases, with decreased oxidative phosphorylation and increased glycolysis [40,57]. The accumulated lactate induces oxidative stress in VSMC and inhibits BNIP3-mediated mitophagy resulting in VSMA apoptosis [58]. The transcription factor NR4A1 is involved in the damage of mitophagy by lactate. In the pathological state, lactate stimulates mitochondrial fission and damages mitochondrial function. Meanwhile, the self-repair mechanism depending on mitophagy is depressed, facilitating VSMC osteoblastic phenotype transition [58]. NR4A1 interrupts the DNA-PKcs/P53 signaling pathway, which is a cellular self-repairing signal that responds to chronic irritation [58]. These studies indicate that mitophagy restores mitochondrial function by removing damaged mitochondria and protects VSMCs from apoptosis. The inhibitory effect of mitophagy on apoptosis is only limited to the early stage of vascular remodeling. When damaged mitochondria accumulate continuously, insufficient mitophagy will accelerate cell apoptosis [59].

Ageing is an independent factor that induces arterial remodeling even in the absence of cardiovascular diseases or other risk factors [60]. Senescent VSMCs are abundant in atherosclerosis plaques and the vascular wall of the artery disease model [42,61]. Defective mitophagy leads to the accumulation of impaired mitochondria, which is the primary source of mtROS and ultimately contributes to the senescence of VSMCs [62]. P53 can bind to PINK and prevent its transport to mitochondria, which leads to impaired mitophagy mediated by PINK/Parkin [63]. It has been shown that activation of PINK/Parkin-mediated mitophagy can alleviate the senescence of VSMCs [62]. In addition to PINK, an AMPK/SIRT1 signal protects mitophagy by improving bioenergy efficiency [63].

The PINK/Parkin and NIX/BNIP3 pathways play significant regulatory roles in the mitophagy of VSMCs. BNIP3 binds to LC3 linkage to activate mitophagy through phosphorylation at Ser17 and Ser24 [64]. In vascular calcification, P53 expression in VSMCs is up-regulated and BINK-mediated mitophagy restriction accelerates VSMC apoptosis [58]. PINK/Parkin is another important pathway for mitophagy in VSMCs. In atherosclerosis, Apelin13 stimulates PINK/Parkin phosphorylation. In response to this signal, AMPK activation inhibits mTOR to promote mitophagy, which protects VSMCs from abnormal proliferation [54]. As a self-repairing process, mitophagy is closely involved in the senescence and apoptosis of VSMCs during vascular remodeling.

## 5. Mitochondrial Genes and Vascular Remodeling

mtDNAs are located in the mitochondrial matrix and encode the proteins of the electron transport chain, including subunits of complexes I, III and IV and ATP synthase [26]. Compared with nuclear DNA, mtDNA is more vulnerable to oxidative stress due to the lack of histone protection and mtROS [65]. mtDNA mutations promote cellular ageing through mitochondrial respiratory chain deficiency and increased production of ROS, which, in turn, accelerates mtDNA mutagenesis [66]. VSMC senescence, elevated ROS levels and accumulated mtDNA damage are the main characteristics of VSMCs in atherosclerotic plaques [67].

Accumulated mtDNA mutation induces age-related phenotypes in both rodent and human cases [68]. Although mtDNA encodes critical proteins in ETC, high levels of pathogenic mtDNA mutation can reduce the ATP and increase mtROS. A decreased ATP level in VSMCs leads to the degradation of the extracellular matrix and decreased collagen synthesis, which increase plaque instability in atherosclerosis [69]. In turn, high level of ROS cause damage in mtDNA and accelerate VSMC apoptosis [58]. ROS and oxidized low-density lipoprotein (ox-LDL) induce mtDNA damage and promote mitophagy without compensatory mitochondriogenesis in plaque VSMCs [70]. In fact, a certain degree of ROS can promote the proliferation of VSMC, but excessive ROS will accelerate the apoptosis of VSMC [39]. mtDNA integrity and copy number are associated with vascular remodeling. Overexpression of the mtDNA helicase Twinkle can improve the mitochondrial respiration of VSMCs independent of ROS and results in a decreased necrotic core and increased stability of the atherosclerosis plaque [70,71].

Aberrant methylation of mtDNA in smooth muscle cells inhibits mtDNA expression and causes mitochondrial respiration defects, which leads to the decreased ATP-dependent contractile function of VSMCs; thus, the progression of vascular stenosis occlusive diseases such as atherosclerosis and post-injury restenosis is exacerbated [72]. Platelet-derived growth factor-BB (PDGF-BB) can evoke DNA methyltransferase 1 (DNMT1)-mediated aberrant methylation of mtDNA in VSMC [72]. These studies suggest that mtDNA transcriptional regulation influences the phenotypic transition of VSMCs and participates in the pathological process (Figure 4).

## 6. VSMC Mitochondria-Targeted Therapy in Vascular Remodeling

In vascular remodeling, the quality control and function of mitochondria are impaired. The therapies focus on restoring the normal function of mitochondria, including OXPHOS, apoptosis and clearing damaged mitochondria. Mitochondrial-targeted antioxidants can scavenge the ROS and stabilize the mitochondrial redox state. Some molecules are designed to specifically target the core pathways and enzymes to affect mitochondrial homeostasis. Adeno-associated virus-2 (AAV-2), modified by peptides, can enhance the specific transgene expression of SMC [73]. During different stages or types of pathological remodeling, the impaired mitochondrial function of VSMCs is different and requires corresponding treatments.

### 6.1. Mitochondrial Function Restoration

Existing evidence has suggested that restoring VSMC mitochondrial function can effectively alleviate the pathological process of vascular remodeling [40,44,50], which could be a promising therapy in a variety of cardiovascular diseases. In rats with pulmonary hypertension, transplantation of healthy mitochondria can restore the contractile phenotype of VSMCs in the pulmonary artery, thereby improving the vasoreactivity and right ventricular remodeling [74]. Superoxide dismutase 2 (SOD2) preferentially localizes to the mitochondria and is endowed with the ability of mtROS clearance. Simvastatin can inhibit the expression of NOR-1, a negative regulatory molecule of SOD2. Therefore, statin is considered to be a potential drug for preventing vascular calcification and reducing mtDNA damage (Table 1) [75]. Elevated mitochondrial oxidative stress is involved in a variety of cardiovascular diseases [76]. Thus, mitochondrial antioxidants are potent agents for protecting VSMC by eliminating excess ROS. Mitochondria-targeted antioxidants MitoQ and astaxanthin (ATX) [77] could attenuate vascular remodeling by inhibiting VSMC phenotype transition and improving mitochondrial function [78]. MitoQ10, a mitochondria-targeted ubiquinone, can accumulate within mitochondria and combines with triphenylphosphonium (TPP) to prevent mitochondrial oxidative damage (Table 1) [79]. Another therapeutic target is epigenetic regulation. SOD-mimetic metalloporphyrin Mn (III) tetrakis (4-benzoic acid) porphyrin (MnTBAP) and DNA methyltransferase inhibitor 5-aza-2-deoxycytidine regress experimental pulmonary arterial hypertension. Tissue-specific, epigenetic regulation of SOD2 improves the mitochondrial redox state and reduces the proliferative, apoptosis-resistant SMC obstructing the pulmonary artery [80].

Mitochondria-dependent apoptosis is impaired in vascular remodeling of pulmonary arterial hypertension. Dichloroacetate (DCA) is a mitochondrial enzyme inhibitor that reduces glycolysis and promotes oxidative phosphorylation. DCA-treated pulmonary hypertension rats obtain decreased medial thickness and right ventricular hypertrophy by affecting apoptosis and K channels [81]. The accumulation of HSP90 (heat shock protein 90) in pulmonary artery SMC mitochondria contributes to their proliferation and resistance to apoptosis and promotes vascular remodeling in pulmonary arterial hypertension. Gamitrinib, a mitochondria-targeted HSP90 inhibitor, regulates mitochondrial homeostasis and alleviates vascular remodeling in a rodent PAH model [82].

**Table 1 ijms-24-03483-t001:** Therapeutic drugs target mitochondria for vascular remodeling diseases.

Therapy	Impact on the VSMC	Related Disease	Target	References
Mitochondrial biogenesis				
Resveratrol	VSMC proliferation	Vascular remodeling and arterial stiffness	SIRT1	[81,83,84]
NR	Mitochondrial respiration of VSMC	Inherited and non-inherited aortic aneurysm	Sirtuins	[32,33]
Mitochondrial dynamics				
Exogenous H_2_S	VSMC proliferation	TypeII diabetes	ROS	[40]
Irish	Phenotypic transformation of VSMC	Chronic kidney disease	AMPK/DRP1	[50]
KLF5	Mitochondrial integrity of VSMC	Abdominal aortic aneurysm	MFN1	[44]
Mdivi-1	VSMC proliferation	Pulmonary hypertension	DRP1	[27,28]
Mitophagy				
Astaxanthin	VSMC proliferation	Hypertension	PINK/Parkin	[77]
Astragaloside IV	VSMC senescence	Ageing	PINK/Parkin	[62]
KTP			PINK/Parkin	[85]
MitoQ	Phenotypic transformation of VSMC	Vascular fibrosis	PINK1/Parkin	[78]
MtDNA damage				
MitoQ10	VSMC respiratory chain	Hypertension	ROS	[79]
Statin	VSMC senescence	Vascular calcification	ROS	[75]
Mitochondria-dependent apoptosis				
MnTBAP	VSMC proliferative and apoptosis	Pulmonary hypertension	SOD2	[80]
DCA	VSMC apoptosis	Pulmonary hypertension	Pyruvate dehydrogenase kinase (PDHK)	[86]
Gamitrinib	VSMC apoptosis	Pulmonary hypertension	mtHSP90	[82]

### 6.2. Mitochondrial Biogenesis Regulation

Maintaining mitochondrial homeostasis by generating new mitochondria via mitochondrial biogenesis is critical for vascular health. The administration of the SIRT1 activator resveratrol can ameliorate pathological stimuli-induced SMC proliferation, vascular remodeling and arterial stiffness [83]. Resveratrol downregulates the expression of AngII type 1 receptor in VSMCs, serum AngII level and aortic expression of angiotensin converting enzyme (ACE) in a rodent model [79,82]. The regulation is similar in other cell types; resveratrol enhanced mitochondrial biogenesis and ameliorated Ang II-induced cardiac remodeling in transgenic rats harboring human renin–angiotensin genes (Table 1) [87].

Nicotinamide riboside (NR) (Table 1) is a nicotinamide adenine dinucleotide (NAD) precursor that promotes mitochondrial biogenesis by increasing PGC1α and TFAM expression through Sirtuin1 and other sirtuins [88]. NR raises TFAM levels and increases mitochondrial respiration in murine and human MFS cells (primary SMCs with FBN1 knockdown). In the Marfan mouse model of thoracic aortic aneurysm, NR treatment can normalize aortic function and diameter [32]. In acute aortic aneurysms and lethal ruptures induced by Ang-II, NR treatment has similar effects on aortic ruptures and prevents sudden death [33].

### 6.3. Mitochondrial Dynamics Regulation

Mitochondrial fusion and fission are involved in the proliferation, migration senescence and phenotypic transformation of VSMCs in vascular remodeling. AngII, high-glucose and oxidative stress facilitate the increase of fission and the decrease of fusion, which is involved in PDGF activation and AMPK depression [35]. GTP-hydrolyzing enzymes, including MFN1/2, OPA1 and DRP1, mediate AMPK and PDGF regulation of mitochondrial dynamics [51]. It is feasible to target these enzymes to maintain the mitochondrial dynamics balance. Mdivi-1 disrupts VSMC proliferation by inhibiting DRP1 expression, which resists the exacerbation in disease models such as intimal hyperplasia and pulmonary hypertension [27,28] (Table 1). Mdivi-1-treated VSMCs showed enhanced morphological integrity and decreased ROS levels. In vascular calcification, Irisin acts on the AMPK/DRP1 pathway to inhibit the transformation of VSMCs to the osteogenic phenotype, thus protecting the function and integrity of mitochondria [50]. Exogenous H_2_S also prevents arterial hyperplasia by inhibiting mitochondrial fission from disrupting VSMC proliferation (Table 1) [40]. Furthermore, the transcription factor KLF5 has been shown to bind to the elF5a promoter to activate its transcription, which can combine with MFN1 to protect mitochondrial integrity (Table 1) [44].

### 6.4. Mitophagy Regulation

Mitophagy in VSMCs is mainly mediated through the PINK/Parkin and BNIP3/NIX pathways [52]. In essence, mitophagy is a self-protective action to remove damaged mitochondria that is involved in the proliferation, senescence and apoptosis of VSMCs [54,75,77]. For atherosclerosis, mitophagy and mitochondrial fission participate in VSMC proliferation together, in which mitochondrial fission dominates [25,59]. Mitochondrial antioxidant astaxanthin can reduce mitochondrial fission and activate mitophagy to decrease the excessive proliferation of VSMCs through PINK/Parkin (Table 1) [77]. In fact, the PINK/Parkin pathway can be activated by a variety of drugs to restore a high-quality mitochondrial status. For example, kinetin triphosphate (KTP) is an ATP analogue and has more affinity with PINK than ATP. This feature of KTP enhances the activity of specific PINK kinases and initiates mitophagy (Table 1) [85]. Mitophagy plays a major role in the senescence of smooth muscle cells. In VSMCs, ROS stimulates the upregulation of P53, which binds to Parkin in the cytoplasm and prevents its mitochondrial localization. This causes mitophagy defects and promotes the senescence of VSMCs [63]. Astragaloside IV has been proven to alleviate the senescence of VSMCs through activating Parkin-mediated mitophagy, which protects vessels from the lesion (Table 1) [62]. Therefore, targeting PINK/Parkin-mediated mitophagy is a feasible method to rescue senescence in VSMC. A mitophagy-related gene (ATG7) deficiency leads to mitophagy defects and exacerbates the development of atherosclerosis [56]. Lactate depresses BNIP3/NIX-mediated mitophagy via the NR4A1/DNA-PKcs/P53 pathway to promote VSMC apoptosis that aggravates vascular calcification [58]. For vascular remodeling diseases, targeting the PINK/Parkin and BNIP3/NIX pathway of VSMC is a prospective therapy that recovers mitophagy to maintain mitochondrial homeostasis.

### 6.5. mtDNA-Targeted Therapy

When the heterogeneity of mutant mtDNA reaches a certain level, abnormal respiratory chain complexes will influence mitochondrial function and cause diseases. Mitochondrial gene therapy is limited by tools. The guide RNA of CRISPR-Cas9 cannot enter the inner mitochondrial membrane to conduct editing. Heteroplasmic shifting is the main method to treat mtDNA damage (Figure 4). Antigenomic therapy decreases mtDNA damage by inhibiting the replication of mutated mtDNA [65]. For instance, peptide nucleic acid (PNA) can specifically bind to a mitochondrial propeptide sequence or TPP, which disrupts the replication of the mutated mtDNA [89,90]. However, it is doubtful whether this method takes effect in vivo; the modified PNAs are not likely to enter the inner mitochondrial membrane and bind to the target position of mtDNA [69]. Using stem cells with fewer mutations to differentiate can also replace cells with mtDNA damage [91]. Overexpressing Twinkle, the mtDNA helicase, increases mtDNA copy number and reduces plaque necrosis, which proves that maintaining mtDNA integrity is a great way to treat vascular disease [71]. mitoTALENs [92] and mtZFNs [93] have been developed to eliminate the mtDNA mutant load and correct single base mutations. However, the copy number depletion and off-target effects of the editing tools still need to be addressed.

## 7. Conclusions

Phenotypically modified VSMCs play a major role in vascular remodeling. Mitochondrial homeostasis is involved in the phenotypic modulation, proliferation, migration, senescence and apoptosis of VSMCs. Here, we focus on the typical factors affecting mitochondrial homeostasis, including mitochondrial function, mitochondrial fission and fusion, mitophagy and mitochondrial DNA mutation. Mitochondrial dysfunction may be a primary cause of vascular remodeling and is a promising target for new therapies. However, whether these strategies can be used in the clinical treatment of vascular remodeling is worth exploring further.

## Figures and Tables

**Figure 1 ijms-24-03483-f001:**
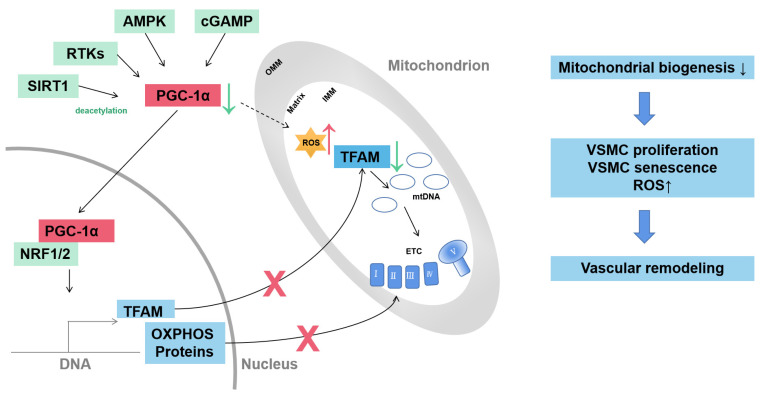
Mitochondrial biogenesis of VSMC in vascular remodeling. Mitochondrial biogenesis is driven by transcriptional coactivator PGC-1α. The expression and activity of PGC-1α are regulated through diverse pathways, including RTKs, cGMP, AMPK and SIRT1. PGC-1α activates transcriptional factors NRF1 and NRF2 to induce transcription of TFAM and nDNA-encoded OXPHOS proteins. TFAM controls the transcription and replication of mtDNA. Decreased PGC-1α contributes to VSMC proliferation, senescence and ROS production and drives vascular remodeling. AMPK, adenosine monophosphate-activated kinase; cGMP, cyclic guanosine monophosphate; ETC, electron transport complex; IMM, inner mitochondrial membrane, mtDNA, mitochondrial DNA; NRF1/2, nuclear respiratory factor-1/2; OMM, outer mitochondrial membrane; OXPHOS, oxidative phosphorylation; RTKs, receptor tyrosine kinases; SIRT1, sirtuin 1; TFAM, mitochondrial transcription factor A; VSMC, vascular smooth muscle cell.

**Figure 2 ijms-24-03483-f002:**
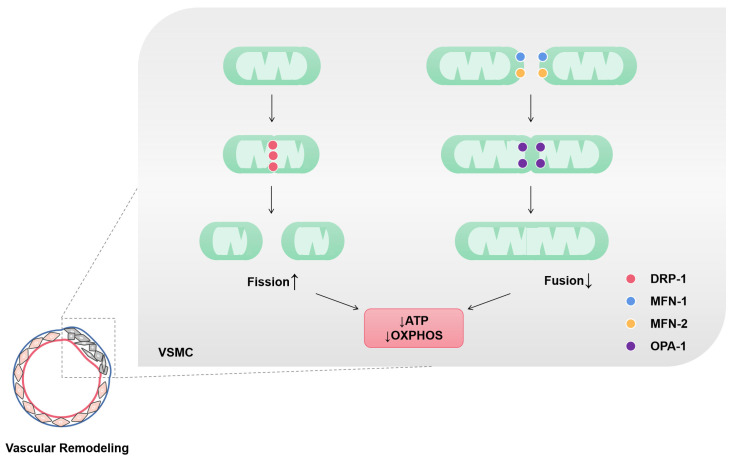
Mitochondrial fission and fusion of VSMCs in vascular remodeling. Mitochondria in vascular remodeling are characterized by increased fission and reduced fusion. Dynamin-related protein-1 (DRP-1) is a GTP hydrolase (GTPase) that initiates fission. Mitochondrial fission is associated with ATP production and O^2^ consumption decrease. Mitofusin-1 and 2 (MFN-1 and MFN-2) are GTPases in the outer mitochondrial membrane (OMM). These two enzymes mediate the fusion of OMM adjacent mitochondria. Optic atrophy protein-1 (OPA-1) is a dynamin-related protein localized in the inner mitochondrial membrane (IMM) that participates in the fusion of IMM. Mitochondrial fusion is associated with an increase in ATP production and OXPHOS.

**Figure 3 ijms-24-03483-f003:**
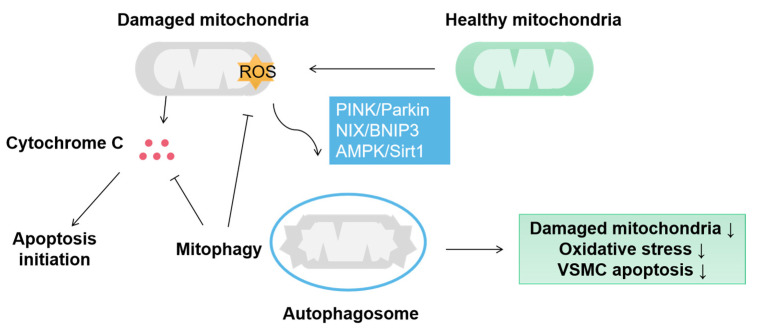
Mitophagy protects VSMC by eliminating damaged mitochondria. In VSMC, mitophagy is mainly aroused by PINK/Parkin, NIX/BNIP3 and AMPK/Sirt1 pathways and induces sequestration of damaged mitochondria by autophagosomes. Mitophagy protects the VSMC from the damage of oxidative stress and cytochrome C-activated apoptosis.

**Figure 4 ijms-24-03483-f004:**
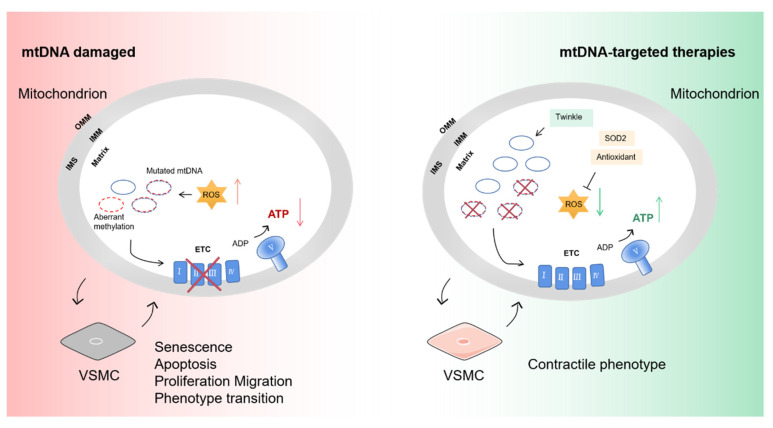
The influence of mtDNA damage on VSMCs and targeted therapies. mtDNA encodes core subunits of respiratory chain complexes I, III and IV and ATP synthase. Pathogenic mutation and aberrant methylation of mtDNA affect oxidative respiratory function. Reduced ATP and increased mtROS drive the senescence of VSMC and phenotype transition from contractile to non-contractile (synthetic). Overexpression of Twinkle increases the mtDNA copy number to shift the heterogeneity level of mutation. Administration of the SOD2 activator and antioxidants can counteract excess mtROS and restore mitochondrial homeostasis. SOD2, superoxide dismutase 2. Part of the elements in this figure are using resources from Servier Medical Art under a Creative Commons Attribution license.

## Data Availability

Not applicable.

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
