# Peer review of "Mitochondrial Homeostasis in VSMCs as a Central Hub in Vascular Remodeling"

_ijms, 2023, doi:10.3390/ijms24043483_

Round 1

Reviewer 1 Report

Type of manuscript: Review

Manuscript ID: ijms-2128578

Title: Mitochondrial Homeostasis in VSMCs as a Central Hub in Vascular Remodeling

In this review Xia and collages try to do a summary of the important mitochondrial process in vascular smoooth muscle cells in vascular remodeling, however the information of the main text does not reflect the entire title.

Major comments:

- Introduction: Vascular remodeling is not only refers to atherosclerosis, there are other pathologies such as aneurym that also are due to a remodeling in the vessel wall. Moreover, the pathophysiology of the vessel wall remodeling either in atherosclerosis or in aneurysm and the importance of the vascular smooth muscle cells in these pathologies is poor, with poor references.

-Authors just speak about fision and fusion, and DNA but nothing is refers to the mitochondrial biogenesis, which is also very important, and even when the title is homeostasis. In relation to this, pathways and gene important in the mitochondrial biogenesis has been related with syndromic vascular remodeling  (Marfan syndrome) ( PMID: 33709773), with AngII dependen disection and aneurysm (PMID: 35196876) and also with restenosis. 

In my opinion the review does not contribute with significant information to the field, and to the previous reviews in the field.

Author Response

In this review Xia and collages try to do a summary of the important mitochondrial process in vascular smoooth muscle cells in vascular remodeling, however the information of the main text does not reflect the entire title.

Major comments:

Comment 1. Introduction: Vascular remodeling is not only refers to atherosclerosis, there are other pathologies such as aneurym that also are due to a remodeling in the vessel wall. Moreover, the pathophysiology of the vessel wall remodeling either in atherosclerosis or in aneurysm and the importance of the vascular smooth muscle cells in these pathologies is poor, with poor references.

 RESPONSE: We sincerely thank the reviewer for this constructive comment. As suggested, we have discussed pathologies of typical vascular diseases, including hypertension, atherosclerosis as well as aneurysm, and added the importance of VSMCs in the pathological processes on the Introduction with updated references. (Manuscript Line 36-53,56-71)

-Authors just speak about fision and fusion, and DNA but nothing is refers to the mitochondrial biogenesis, which is also very important, and even when the title is homeostasis. In relation to this, pathways and gene important in the mitochondrial biogenesis has been related with syndromic vascular remodeling  (Marfan syndrome) ( PMID: 33709773), with AngII dependent dissection and aneurysm (PMID: 35196876) and also with restenosis. 

In my opinion the review does not contribute with significant information to the field, and to the previous reviews in the field.

RESPONSE: We thank the reviewer for this valuable suggestion. We agree with the reviewer and have added the content of important pathways and genes in the mitochondrial biogenesis (Manuscript 2. Line 94-97 and Figure 1 in page 10), mitochondrial biogenesis and vascular remodeling (Manuscript 2. Line 98-108) and the targeted treatments (Manuscript 6.2. Line 258-270). The two suggested references have also been cited. We hope the revised manuscript is now deemed of sufficient merit to be considered for publication.

Reviewer 2 Report

This is a well written review that presents the mitochondria quality/function controlled by fission/fusion homeostasis and mitophagy is critical for VSMC fate, which is tightly associated with vascular remodeling. I make only a few minor suggestions, which I believe may improve the manuscript.

1. Although Fig. 1 title mentioned vascular remodeling, there is no vascular remodeling information in Fig. 1. It is better to provide this information in Fig. 1.

2. For Table 1, it needs to add related vascular remodeling diseases for each therapy, such as atherosclerosis, abdominal aortic aneurysm, hypertension, and neointima formation, etc.

3. Need to discuss how to specifically target VSMC mitochondria quality and function for anti-vascular remodeling.

4. Should complete reference information, including references 17 and 47. Check whether the cited reference matches the text. For example, reference 8.

Author Response

This is a well written review that presents the mitochondria quality/function controlled by fission/fusion homeostasis and mitophagy is critical for VSMC fate, which is tightly associated with vascular remodeling. I make only a few minor suggestions, which I believe may improve the manuscript.

Comment 1. Although Fig. 1 title mentioned vascular remodeling, there is no vascular remodeling information in Fig. 1. It is better to provide this information in Fig. 1.

RESPONSE: Thanks for this valuable suggestion. We have added the information about vascular remodeling in the revised Fig. 1 (Page 11, Figure 2).

Comment 2. For Table 1, it needs to add related vascular remodeling diseases for each therapy, such as atherosclerosis, abdominal aortic aneurysm, hypertension, and neointima formation, etc.

RESPONSE: We appreciate the reviewer for the helpful comment. Related vascular remodeling diseases for each therapy have been inserted in the revised Table 1. (Manuscript Line 312-313).

Comment 3. Need to discuss how to specifically target VSMC mitochondria quality and function for anti-vascular remodeling.

RESPONSE: We thank the reviewer for this constructive comment. Indeed, several antioxidants and molecules have been designed to specifically target mitochondria. AAV vectors was used to deliver genes specifically into SMCs. These tools were used to specifically target VSMC mitochondria to replace damaged mitochondria, improve the ATP production, stabilize mitochondrial redox state and restore apoptosis in anti-vascular remodeling. We have inserted additional discussion into the revised manuscript with updated references (Manuscript Line 224-229 and Manuscript 6.1. Line 232-255).

Comment 4. Should complete reference information, including references 17 and 47. Check whether the cited reference matches the text. For example, reference 8.

RESPONSE: As suggested, we reformatted all the references with complete information, including references 17 and 47 (now the updated references 26 and 62). In addition, we replaced the original reference 8 with a matched reference (now the updated reference 17).

Round 2

Reviewer 1 Report

The authors have improved the revised version, adding part of the references and text that was required in the previous version.

Reviewer 2 Report

The concerns have been addressed. There is no more comment.